# Improved Transformer for High-Resolution GANs

**Long Zhao**[1,*]   **Zizhao Zhang**[2]    **Ting Chen**[3]    **Dimitris N. Metaxas**[1]    **Han Zhang**[3]

[1]Rutgers University    [2]Google Cloud AI    [3]Google Research

## Abstract

Attention-based models, exemplified by the Transformer, can effectively model long range dependency, but suffer from the quadratic complexity of self-attention operation, making them difficult to be adopted for high-resolution image generation based on Generative Adversarial Networks (GANs). In this paper, we introduce two key ingredients to Transformer to address this challenge. First, in low-resolution stages of the generative process, standard global self-attention is replaced with the proposed multi-axis blocked self-attention which allows efficient mixing of local and global attention. Second, in high-resolution stages, we drop self-attention while only keeping multi-layer perceptrons reminiscent of the implicit neural function. To further improve the performance, we introduce an additional self-modulation component based on cross-attention. The resulting model, denoted as HiT, has a nearly linear computational complexity with respect to the image size and thus directly scales to synthesizing high definition images. We show in the experiments that the proposed HiT achieves state-of-the-art FID scores of 30.83 and 2.95 on unconditional ImageNet $128 \times 128$ and FFHQ $256 \times 256$, respectively, with a reasonable throughput. We believe the proposed HiT is an important milestone for generators in GANs which are completely free of convolutions. Our code is made publicly available at `https://github.com/google-research/hit-gan`.

## 1 Introduction

Attention-based models demonstrate notable learning capabilities for both encoder-based and decoder-based architectures [66, 69] due to their self-attention operations which can capture long-range dependencies in data. Recently, Vision Transformer [12], one of the most powerful attention-based models, has achieved a great success on encoder-based vision tasks, specifically image classification [12, 60], segmentation [37, 64], and vision-language modeling [46]. However, applying the Transformer to image generation based on Generative Adversarial Networks (GANs) is still an open problem.

The main challenge of adopting the Transformer as a decoder/generator lies in two aspects. On one hand, the quadratic scaling problem brought by the self-attention operation becomes even worse when generating pixel-level details for high-resolution images. For example, synthesizing a high definition image with the resolution of $1024 \times 1024$ leads to a sequence containing around one million pixels in the final stage, which is unaffordable for the standard self-attention mechanism. On the other hand, generating images from noise inputs poses a higher demand for spatial coherency in structure, color, and texture than discriminative tasks, and hence a more powerful yet efficient self-attention mechanism is desired for decoding feature representations from inputs.

In view of these two key challenges, we propose a novel Transformer-based decoder/generator in GANs for high-resolution image generation, denoted as HiT. HiT employs a hierarchical structure of Transformers and divides the generative process into low-resolution and high-resolution stages, focusing on feature decoding and pixel-level generating, respectively. Specifically, its low-resolution

---

*This work was done while Long Zhao was a student researcher at the Google Brain team. Correspondence to: Long Zhao (lz311@rutgers.edu) and Han Zhang (zhanghan@google.com).

35th Conference on Neural Information Processing Systems (NeurIPS 2021).

stages follow the design of Nested Transformers [73] but enhanced by the proposed multi-axis blocked self-attention to better capture global information. Assuming that spatial features are well decoded after low-resolution stages, in high-resolution stages, we drop all self-attention operations in order to handle extremely long sequences for high definition image generation. The resulting high-resolution stages of HiT are built by multi-layer perceptrons (MLPs) which have a linear complexity with respect to the sequence length. Note that this design aligns with the recent findings [58, 59] that pure MLPs manage to learn favorable features for images, but it simply reduces to an implicit neural function [41, 42, 43] in the case of generative modeling. To further improve the performance, we present an additional cross-attention module that acts as a form of self-modulation [8]. In summary, this paper makes the following contributions:

- We propose HiT, a Transformer-based generator for high-fidelity image generation. Standard self-attention operations are removed in the high-resolution stages of HiT, reducing them to an implicit neural function. The resulting architecture easily scales to high definition image synthesis (with the resolution of $1024 \times 1024$) and has a comparable throughput to StyleGAN2 [28].

- To tame the quadratic complexity and enhance the representation capability of self-attention operation in low-resolution stages, we present a new form of sparse self-attention operation, namely multi-axis blocked self-attention. It captures local and global dependencies within non-overlapping image blocks in parallel by attending to a single axis of the input tensor at a time, each of which uses a half of attention heads. The proposed multi-axis blocked self-attention is efficient, simple to implement, and yields better performance than other popular self-attention operations [37, 62, 73] working on image blocks for generative tasks.

- In addition, we introduce a cross-attention module performing attention between the input and intermediate features. This module re-weights intermediate features of the model as a function of the input, playing a role as self-modulation [8], and provides important global information to high-resolution stages where self-attention operations are absent.

- The proposed HiT obtains competitive FID [16] scores of 30.83 and 2.95 on unconditional ImageNet [49] $128 \times 128$ and FFHQ [28] $256 \times 256$, respectively, highly reducing the gap between ConvNet-based GANs and Transformer-based ones. We also show that HiT not only works for GANs but also can serve as a general decoder for other models such as VQ-VAE [61]. Moreover, we observe that the proposed HiT can obtain more performance improvement from regularization than its ConvNet-based counterparts. To the best of our knowledge, these are the best reported scores for an image generator that is completely free of convolutions, which is an important milestone towards adopting Transformers for high-resolution generative modeling.

## 2 Related work

**Transformers for image generation.** There are two main streams of image generation models built on Transformers [63] in the literature. One stream of them [14, 44] is inspired by auto-regressive models that learn the joint probability of the image pixels. The other stream focuses on designing Transformer-based architecture for generative adversarial networks (GANs) [15]. This work follows the spirit of the second stream. GANs have made great progress in various image generation tasks, such as image-to-image translation [57, 74, 75, 79] and text-to-image synthesis [70, 71], but most of them depend on ConvNet-based backbones. Recent attempts [23, 32] build a pure Transformer-based GAN by a careful design of attention hyper-parameters as well as upsampling layers. However, such a model is only validated on small scale datasets (e.g., CIFAR-10 [31] and STL-10 [10] consisting of $32 \times 32$ images) and does not scale to complex real-world data. To our best knowledge, no existing work has successfully applied a Transformer-based architecture completely free of convolutions for high-resolution image generation in GANs.

GANformer [21] leverages the Transformer as a plugin component to build the bipartite structure to allow long-range interactions during the generative process, but its main backbone is still a ConvNet based on StyleGAN [28]. GANformer and HiT are different in the goal of using attention modules. GANformer utilizes the attention mechanism to model the dependences of objects in a generated scene/image. Instead, HiT explores building efficient attention modules for synthesizing general objects. Our experiments demonstrate that even for simple face image datasets, incorporating the attention mechanism can still lead to performance improvement. We believe our work reconfirms the necessity of using attention modules for general image generation tasks, and more importantly, we

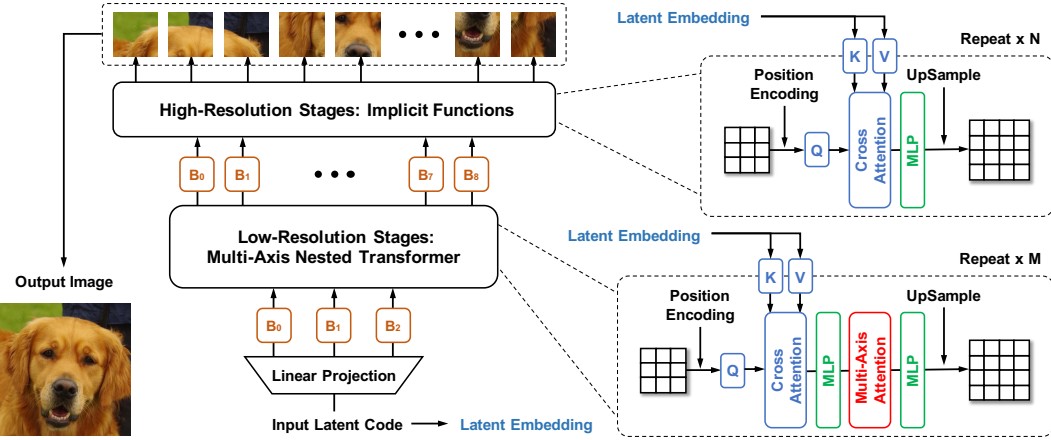

Figure 1: **HiT architecture**. In each stage, input the feature is first organized into blocks (denoted as $B_i$). HiT's low-resolution stages follow the decoder design of Nested Transformer [73] which is then enhanced by the proposed multi-axis blocked self-attention. We drop self-attention operations in the high-resolution stages, resulting in implicit neural functions. The model is further boosted by cross-attention modules which allow intermediate features to be modulated directly by the input latent code. The detailed algorithm can be found in the supplementary materials.

present an efficient architecture for attention-based high-resolution generators which might benefit the design of future attention-based GANs.

**Attention models.** Many efforts [2, 17, 65] have been made to reduce the quadratic complexity of self-attention operation. When handling vision tasks, some works [37, 62, 73] propose to compute local self-attention in blocked images which takes advantage of the grid structure of images. [6] also presents local self-attention within image blocks but it does not perform self-attention across blocks as in HiT. The most related works to the proposed multi-axis blocked self-attention are [17, 65] where they also compute attention along axes. But our work differs notably in that we compute different attentions within heads on blocked images. Some other works [7, 21, 22] avoid directly applying standard self-attention to the input pixels, and perform attention between the input and a small set of latent units. Our cross-attention module differs from them in that we apply cross-attention to generative modeling designed for pure Transformer-based architectures.

**Implicit neural representations.** The largest popularity of implicit neural representations/functions (INRs) is studied in 3D deep learning to represent a 3D shape in a cheap and continuous way [41, 42, 43]. Recent studies [1, 13, 34, 54] explore the idea of using INRs for image generation, where they learn a hyper MLP network to predict an RGB pixel value given its coordinates on the image grid. Among them, [1, 54] are closely related to our high-resolution stages of the generative process. One remarkable difference is that our model is driven by the cross-attention module and features generated in previous stages instead of the hyper-network presented in [1, 54].

## 3 Approach

### 3.1 Main Architecture

In the case of unconditional image generation, HiT takes a latent code $z \sim \mathcal{N}(\mathbf{0}, \boldsymbol{I})$ as input and generates an image of the target resolution through a hierarchical structure. The latent code is first projected into an initial feature with the spatial dimension of $H_0 \times W_0$ and channel dimension of $C_0$. During the generation process, we gradually increase the spatial dimension of the feature map while reducing the channel dimension in multiple stages. We divide the generation stages into low-resolution stages and high-resolution stages to balance feature dependency range in decoding and computation efficiency. The overview of the proposed method is illustrated in Figure 1.

In low-resolution stages, we allow spatial mixing of information by efficient attention mechanism. We follow the decoder form of Nested Transformer [73] where in each stage, the input feature is first

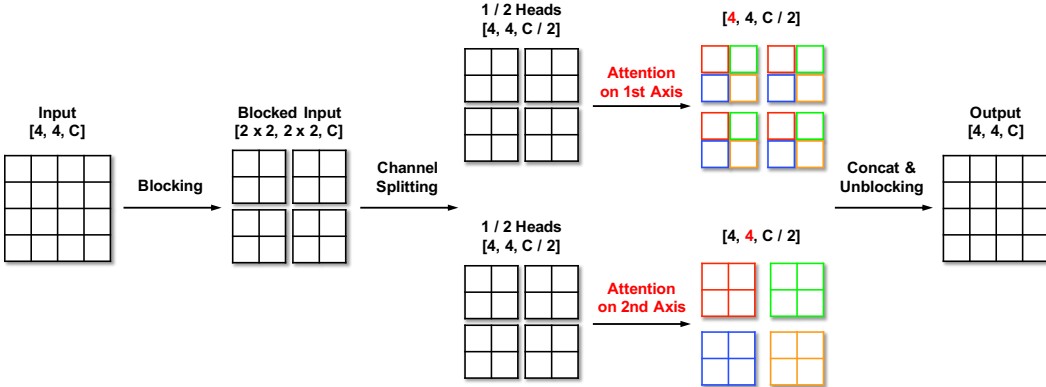

Figure 2: **Multi-axis self-attention architecture.** The different stages of multi-axis self-attention for a $[4, 4, C]$ input with the block size of $b = 2$. The input is first blocked into $2 \times 2$ non-overlapping $[2, 2, C]$ patches. Then regional and dilated self-attention operations are computed along two different axes, respectively, each of which uses a half of attention heads. The attention operations run in parallel for each of the tokens and their corresponding attention regions, illustrated with different colors. The spatial dimensions after attention are the same as the input image.

divided into non-overlapping blocks where each block can be considered as a local patch. After being combined with learnable positional encoding, each block is processed independently via a shared attention module. We enhance the local self-attention of Nested Transformer by the proposed multi-axis blocked self-attention that can produce a richer feature representation by explicitly considering local (within blocks) as well as global (across blocks) relations. We denote the overall architecture of these stages as multi-axis Nested Transformer.

Assuming that spatial dependency is well modeled in the low-resolution stages, the high-resolution stages can focus on synthesizing pixel-level image details purely based on the local features. Thus in the high-resolution stage, we remove all self-attention modules and only maintain MLPs which can further reduce computation complexity. The resulting architecture in this stage can be viewed as an implicit neural function conditioned on the given latent feature as well as positional information.

We further enhance HiT by adding a cross-attention module at the beginning of each stage which allows the network to directly condition the intermediate features on the initial input latent code. This kind of self-modulation layer leads to improved generative performance, especially when self-attention modules are absent in high-resolution stages. In the following sections, we provided detailed descriptions for the two main architectural components of HiT: (i) multi-axis blocked self-attention module and (ii) cross-attention module, respectively.

## 3.2 Multi-Axis Blocked Self-Attention

Different from the blocked self-attention in Nested Transformer [73], the proposed multi-axis blocked self-attention performs attention on more than a single axis. The attentions performed in two axes correspond to two forms of sparse self-attention, namely regional attention and dilated attention. Regional attention follows the spirit of blocked local self-attention [37, 62] where tokens attend to their neighbours within non-overlapped blocks. To remedy the loss of global attention, dilated attention captures long-range dependencies between tokens across blocks: it subsamples attention regions in a manner similar to dilated convolutions with a fixed stride equal to the block size. Figure 2 illustrates an example of these two attentions.

To be specific, given an input of image with size $(H, W, C)$, it is blocked into a tensor $X$ of the shape $(b \times b, \frac{H}{b} \times \frac{W}{b}, C)$ representing $(b, b)$ non-overlapping blocks each with the size of $(\frac{H}{b}, \frac{W}{b})$. Dilated attention mixes information along the first axis of $X$ while keeping information along other axes independent; regional attention works in an analogous manner over the second axis of $X$. They are straightforward to implement: attention over the $i$-th axis of $X$ can be implemented by einsum operation which is available in most deep learning libraries.

We mix regional and dilated attention in a single layer by computing them in parallel, each of which uses a half of attention heads. Our method can be easily extended to model more than two axes by performing attention on each of the axis in parallel. Axial attention [17, 20, 65] can be viewed as a special case of our method, where the blocking operation before attention is removed. However, we find blocking is the key to achieve significant performance improvement in the experiment. This is because the blocked structure reveals a better inductive bias for images. Compared with [17, 37, 65] where different attention modules are interleaved in consecutive layers, our approach aggregates local and global information in a single round, which is not only more flexible for architecture design but also shown to yield better performance than interleaving in our experiment.

**Balancing attention between axes.** In each multi-axis blocked self-attention module, the input feature is blocked in a balanced way such that we have $b \times b \approx \frac{H}{b} \times \frac{W}{b}$. This ensures that regional and dilated attention is computed on an input sequence of a similar length, avoiding half of the attention heads are attended to a too sparse region. In general, performing dot-product attention between two input sequences of length $N = H \times W$ requires a total of $\mathcal{O}(N^2)$ computation. When computing the balanced multi-axis blocked attention on an image with the block size of $S$, i.e., $S = \sqrt{N}$, we perform attention on $S$ sequences of length $S$, which is a total of $\mathcal{O}(S \cdot S^2) = \mathcal{O}(N\sqrt{N})$ computation. This leads to an $\mathcal{O}(\sqrt{N})$ saving in computation over standard self-attention.

### 3.3 Cross-Attention for Self-Modulation

To further improve the global information flow, we propose to let the intermediate features of the model directly attend to a small tensor projected from the input latent code. This is implemented via a cross-attention operation and can be viewed as a form of self-modulation [8]. The proposed technique has two benefits. First, as shown in [8], self-modulation stabilizes the generator towards favorable conditioning values and also appears to improve mode coverage. Second, when self-attention modules are absent in high-resolution stages, attending to the input latent code provides an alternative way to capture global information when generating pixel-level details.

Formally, let $X_l$ be the first-layer feature representation of the $l$-th stage. The input latent code $z$ is first projected into a 2D spatial embedding $Z$ with the resolution of $H_Z \times W_Z$ and dimension of $C_Z$ by a linear function. $X_l$ is then treated as the query and $Z$ as the key and value. We compute their cross-attention following the update rule: $X_l' = \text{MHA}(X_l, Z + P_Z)$, where MHA represents the standard mulit-head self-attention, $X_l'$ is the output, and $P_Z$ is the learnable positional encoding having the same shape as $Z$. Note that $Z$ is shared across all stages. For an input feature with the sequence length of $N$, the embedding size is a pre-defined hyperparameter far less than $N$ (i.e., $H_Z \times W_Z \ll N$). Therefore, the resulting cross-attention operation has linear complexity $\mathcal{O}(N)$.

In our initial experiments, we find that compared with cross-attention, using AdaIN [19] and modulated layers [28] for Transformer-based generators requires much higher memory cost during model training, which usually leads to out-of-memory errors when the model is trained for generating high-resolution images. As a result, related work like ViT-GAN [32], which uses AdaIN and modulated layers, can only produce images up to the resolution of $64 \times 64$. Hence, we believe cross-attention is a better choice for high-resolution generators based on Transformers.

## 4 Experiments

### 4.1 Experiment Setup

**Datasets.** We validate the proposed method on three datasets: ImageNet [49], CelebA-HQ [25], and FFHQ [28]. ImageNet (LSVRC2012) [49] contains roughly 1.2 million images with 1000 distinct categories and we down-sample the images to $128 \times 128$ and $256 \times 256$ by bicubic interpolation. We use random crop for training and center crop for testing. This dataset is challenging for image generation since it contains samples with diverse object categories and textures. We also adopt ImageNet as the main test bed during the ablation study.

CelebA-HQ [25] is a high-quality version of the CelebA dataset [38] containing 30,000 of the facial images at $1024 \times 1024$ resolution. To align with [25], we use these images for both training and evaluation. FFHQ [28] includes vastly more variation than CelebA-HQ in terms of age, ethnicity and image background, and also has much better coverage of accessories such as eyeglasses, sunglasses,

and hats. This dataset consists of 70,000 high-quality images at $1024 \times 1024$ resolution, out of which we use 50,000 images for testing and train models with all images following [28]. We synthesize images on these two datasets with the resolutions of $256 \times 256$ and $1024 \times 1024$.

**Evaluation metrics.** We adopt the Inception Score (IS) [51] and the Fréchet Inception Distance (FID) [16] for quantitative evaluation. Both metrics are calculated based on a pre-trained Inception-v3 image classifier [56]. Inception score computes KL-divergence between the real image distribution and the generated image distribution given the pre-trained classifier. Higher inception scores mean better image quality. FID is a more principled and comprehensive metric, and has been shown to be more consistent with human judgments of realism [16, 70]. Lower FID values indicate closer distances between synthetic and real data distributions. In our experiments, 50,000 samples are randomly generated for each model to calculate the inception score and FID on ImageNet and FFHQ, while 30,000 samples are produced for comparison on CelebA-HQ. Note that we follow [51] to split the synthetic images into groups (5000 images per group) and report their averaged inception score.

## 4.2 Implementation Details

**Architecture configuration.** In our implementation, HiT starts from an initial feature of size $8 \times 8$ projected from the input latent code. We use pixel shuffle [53] for upsampling the output of each stage, as we find using nearest neighbors leads to model failure which aligns with the observation in Nested Transformer [73]. The number of low-resolution stages is fixed to be 4 as a good trade-off between computational speed and generative performance. For generating images larger than the resolution of $128 \times 128$, we scale HiT to different model capacities – small, base, and large, denoted as HiT-{S, B, L}. We refer to the supplementary materials for their detailed architectures.

It is worth emphasizing that the flexibility of the proposed multi-axis blocked self-attention makes it possible for us to build smaller models than [37]. This is because they interleave different types of attention operations and thus require the number of attention blocks in a single stage to be even (at least 2). In contrast, our multi-axis blocked self-attention combines different attention outputs within heads which allows us to use an arbitrary number (e.g., 1) of attention blocks in a model.

**Training details.** In all the experiments, we use a ResNet-based discriminator following the architecture design of [28]. Our model is trained with a standard non-saturating logistic GAN loss with $R_1$ gradient penalty [40] applied to the discriminator. $R_1$ penalty penalizes the discriminator for deviating from the Nash-equilibrium by penalizing the gradient on real data alone. The gradient penalty weight is set to 10. Adam [29] is utilized for optimization with $\beta_1 = 0$ and $\beta_2 = 0.99$. The learning rate is 0.0001 for both the generator and discriminator. All the models are trained using TPU for one million iterations on ImageNet and 500,000 iterations on FFHQ and CelebA-HQ. We set the mini-batch size to 256 for the image resolution of $128 \times 128$ and $256 \times 256$ while to 32 for the resolution of $1024 \times 1024$. To keep our setup simple, we do not employ any training tricks for GAN training, such as progressive growing, equalized learning rates, pixel normalization, noise injection, and mini-batch standard deviation that recent literature demonstrate to be cruicial for obtaining state-of-the-art results [25, 28]. When training on FFHQ and CelebA-HQ, we utilize balanced consistency regularization (bCR) [72, 77] for additional regularization where images are augmented by flipping, color, translation, and cutout as in [76]. bCR enforces that for both real and generated images, two sets of augmentations applied to the same input image should yield the same output. bCR is only added to the discriminator with the weight of 10. We also decrease the learning rate by half to stabilize the training process when bCR is used.

## 4.3 Results on ImageNet

**Unconditional generation.** We start by evaluating the proposed HiT on the ImageNet $128 \times 128$ dataset, targeting the unconditional image generation setting for simplicity. In addition to recently reported state-of-the-art GANs, we implement a ConvNet-based generator following the widely-used architecture from [69] while using the exactly same training setup of the proposed HiT (e.g., losses and $R_1$ gradient penalty) denoted as ConvNet-$R_1$ for a fair comparison. The results are shown in Table 1. We can see that HiT outperforms the previous ConvNet-based methods by a notable margin in terms of both IS and FID scores. Note that as reported in [11], BigGAN [4] achieves 30.91 FID on unconditional ImageNet, but its model is far larger (more than 70M parameters) than HiT (around 30M parameters). Therefore, the results are not directly comparable. Even though, HiT (30.83 FID)

Table 1: Comparison with the state-of-the-art methods on the ImageNet $128 \times 128$ dataset. $^{\dagger}$ is based on a supervised pre-trained ImageNet classifier.

| Method | FID ↓ | IS ↑ |
|---|---|---|
| Vanilla GAN [15] | 54.17 | 14.01 |
| PacGAN2 [35] | 57.51 | 13.50 |
| MGAN [18] | 58.88 | 13.22 |
| Logo-GAN-AE [50] | 50.90 | 14.44 |
| Logo-GAN-RC [50]$^{\dagger}$ | 38.41 | 18.86 |
| SS-GAN (sBN) [9] | 43.87 | - |
| Self-Conditioned GAN [36] | 40.30 | 15.82 |
| ConvNet-$R_1$ | 37.18 | 19.55 |
| HiT (Ours) | **30.83** | **21.64** |

Table 2: Reconstruction FID on the ImageNet $256 \times 256$ dataset. We note that VQ-VAE-2 utilizes a hierarchical organization of VQ-VAE and thus has two codebooks $\mathcal{Z}$.

| Method | Embedding size and $|\mathcal{Z}|$ | FID ↓ |
|---|---|---|
| VQ-VAE [61] | 32, 1024 | 75.19 |
| DALL-E [47] | 32, 8192 | 34.30 |
| VQ-VAE-2 [48] | 64, 512
32, 512 | 10.00 |
| VQGAN [14] | 16, 1024 | 8.00 |
| VQ-HiT (Ours) | 16, 1024 | **6.37** |

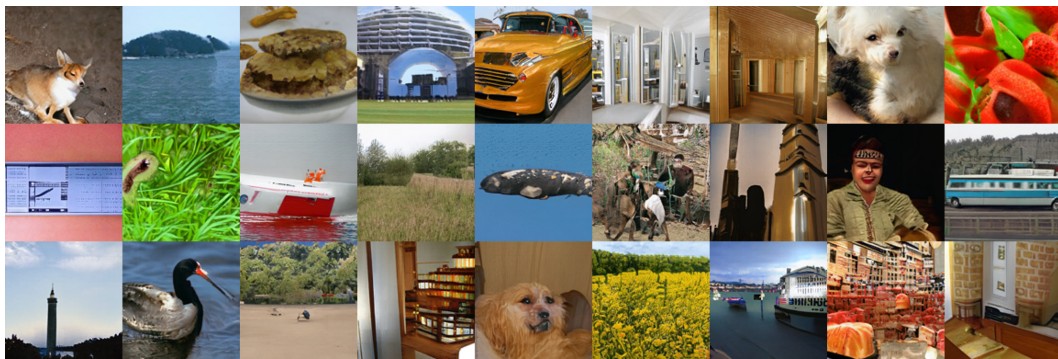

Figure 3: Unconditional image generation results of HiT trained on ImageNet $128 \times 128$.

is slightly better than BigGAN and achieves the state of the art among models with a similar capacity as shown in Table 1. We also note that [8, 36] leverage auxiliary regularization techniques and our method is complementary to them. Examples generated by HiT on ImageNet are shown in Figure 3, from which we observe nature visual details and diversity.

**Reconstruction.** We are also interested in the reconstruction ability of the proposed HiT and evaluate by employing it as a decoder for the vector quantised variational auto-encoder (VQ-VAE [61]), a state-of-the-art approach for visual representation learning. In addition to the reconstruction loss and adversarial loss, our HiT-based VQ-VAE variant, namely VQ-HiT, is also trained with the perceptual loss [24] following the setup of [14, 78]. Please refer to the supplementary materials for more details on the architecture design and model training. We evaluate the metric of reconstruction FID on ImageNet $256 \times 256$ and report the results in Table 2. Our VQ-HiT attains the best performance while providing significantly more compression (i.e., smaller embedding size and fewer number of codes in the codebook $\mathcal{Z}$) than [47, 48, 61].

## 4.4 Higher Resolution Generation

**Baselines.** To demonstrate the utility of our approach for high-fidelity images, we compare to state-of-the-art techniques for generating images on CelebA-HQ and FFHQ, focusing on two common resolutions of $256 \times 256$ and $1024 \times 1024$. The main competing method of the proposed HiT is StyleGAN [27, 28] – a hypernetwork-based ConvNet archieving the best performance on these two datasets. On the FFHQ dataset, apart from our ConvNet-based counterparts, we also compare to the most recent INR-based methods including CIPS [1] and INR-GAN [54]. These two models are closely related to HiT as the high-resolution stages of our approach can be viewed as a form of INR. Following the protocol of [1], we report the results of StyleGAN2 [28] trained without style-mixing and path regularization of the generator on this dataset.

Table 3: Comparison with the state-of-the-art methods on CelebA-HQ (**left**) and FFHQ (**right**) with the resolutions of $256 \times 256$ and $1024 \times 1024$. $\star$ bCR is not applied at the $1024 \times 1024$ resolution.

| Method | FID ↓ (CelebA-HQ) $\times 256$ | $\times 1024$ |
|---|---|---|
| VAEBM [68] | 20.38 | - |
| StyleALAE [45] | 19.21 | - |
| PG-GAN [25] | 8.03 | - |
| COCO-GAN [33] | - | 9.49 |
| VQGAN [14] | 10.70 | - |
| StyleGAN [27] | - | **5.17** |
| HiT-B (Ours) | **3.39** | 8.83$^\star$ |

| Method | FID ↓ (FFHQ) $\times 256$ | $\times 1024$ |
|---|---|---|
| U-Net GAN [52] | 7.63 | - |
| StyleALAE [45] | - | 13.09 |
| VQGAN [14] | 11.40 | - |
| INR-GAN [54] | 9.57 | 16.32 |
| CIPS [1] | 4.38 | 10.07 |
| StyleGAN2 [28] | 3.83 | **4.41** |
| HiT-B (Ours) | **2.95** | 6.37$^\star$ |

Table 4: Comparison with the main competing methods in terms of number of network parameters, throughput, and FID on FFHQ $256 \times 256$. The throughput is measured on a single Tesla V100 GPU.

| Architecture | Model | #params (million) | Throughput (images / sec) | FID ↓ (FFHQ $\times 256$) |
|---|---|---|---|---|
| ConvNet | StyleGAN2 [28] | 30.03 | 95.79 | 3.83 |
| INR | CIPS [1] | 45.90 | 27.27 | 4.38 |
| | INR-GAN [54] | 107.03 | 266.45 | 9.57 |
| Transformer | HiT-S | 38.01 | 86.64 | 3.06 |
| | HiT-B | 46.22 | 52.09 | 2.95 |
| | HiT-L | 97.46 | 20.67 | 2.58 |

**Results.** We report the results in Table 3. Impressively, the proposed HiT obtains the best FID scores at the resolution of $256 \times 256$ and sets the new state of the art on both datasets. Meanwhile, our performance is also competitive at the resolution of $1024 \times 1024$ but only slightly shy of StyleGAN. This is due to our finding that conventional regularization techniques such as [77] cannot be directly applied to Transformer-based architectures for synthesizing ultra high-resolution images. We believe this triggers a new research direction and will explore it as our future work. It is also worth mentioning that our method consistently outperforms INR-based models, which suggests the importance of involving the self-attention mechanism into image generation.

Table 4 provides a more detailed comparison to our main competing methods in terms of the number of parameters, throughput, and FID on FFHQ $256 \times 256$. We find that the proposed HiT-S has a comparable runtime performance (throughput) with StyleGAN while yielding better generative results. More importantly, FID scores can be further improved when larger variants of HiT are employed. Figure 4 illustrates our synthetic face images on CelebA-HQ.

## 4.5 Ablation Study

We then evaluate the importance of different components of our model by its ablation on ImageNet $128 \times 128$. A lighter-weight training configuration is employed in this study to reduce the training time. We start with a baseline architecture without any attention modules which reduces to a pure implicit neural function conditioned on the input latent code [3, 30]. We build upon this baseline by gradually incorporating cross-attention and self-attention modules. We compare the methods that perform attention without blocking (including standard self-attention [63], axial attention [17, 65]) and with blocking (including blocked local attention [62, 73]). We also compare a variant of our method where different types of attention are interleaved other than combined in attention heads.

The results are reported in Table 5. As we can see, the latent code conditional INR baseline cannot fit training data favourably since it lacks an efficient way to capture the latent code during the generative process. Interestingly, incorporating the proposed cross-attention module improves the baseline and achieves the state of the art, thanks to its function of self-modulation. More importantly, we find that blocking is vital for attention: performing different types of attention after blocking can all improve the performance by a notable margin while the axial attention cannot. This is due to the fact that the

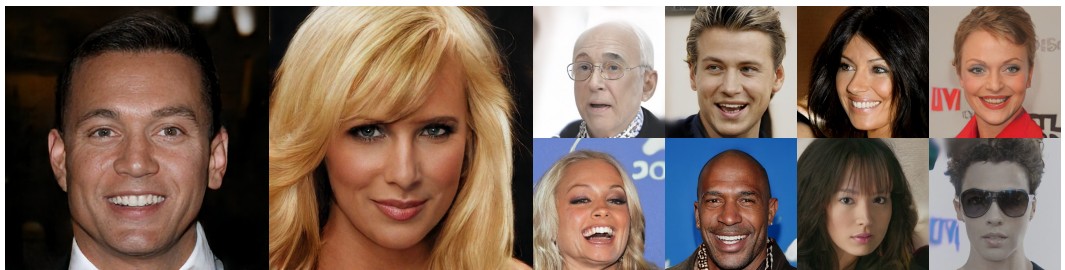

Figure 4: Synthetic face images by HiT-B trained on CelebA-HQ $1024 \times 1024$ and $256 \times 256$.

Table 5: Ablation study. We start with the INR-based generator [3, 30] conditioned on the input latent code and gradually improve it with the proposed attention components and their variations. O/M denotes "out-of-memory" error: the model cannot be trained for the batch size of one.

| | Model configuration | #params (million) | Throughput (images / sec) | FID ↓ | IS ↑ |
|---|---|---|---|---|---|
| | Latent-code conditioned INR decoder [3, 30] | 42.68 | 110.39 | 56.33 | 16.19 |
| + | Cross-attention for self-modulation | 61.55 | 82.67 | 35.94 | 19.42 |
| one of | All-to-all self-attention [63] | 67.60 | - | O/M | O/M |
| | Axial attention [17, 20, 65] | 67.60 | 74.21 | 35.15 | 19.79 |
| + | Blocked local attention [62, 73] Interleaving blocked regional and dilated attention Multi-axis blocked self-attention (Ours) | 67.60 | 75.54 | 33.70 32.96 32.23 | 19.96 20.75 20.96 |
| + | Balancing attention between axes (Full model) | 67.60 | 75.33 | **31.87** | **21.32** |

image structure after blocking introduces a better inductive bias for images. At last, we observe that the proposed multi-axis blocked self-attention yields better FID scores than interleaving different attention outputs, and it achieves the best performance after balancing the attention axes.

In Table 6, we investigate another form of our variations, where we incorporate attention in different number of stages across the generator. We can see that the more in the stack the attention is applied (low-resolution stages), the better the model performance will be. This study provides a validation for the effectiveness of self-attention operation in generative modeling. However, for feature resolutions larger than $64 \times 64$, we do not observe obvious performance gains brought by self-attention operations but a notable degradation in both training and inference time.

### 4.6 Effectiveness of Regularization

We further explore the effectiveness of regularization for the proposed Transformer-based generator. Note that different from the recent interest in training GANs in the few-shot scenarios [26, 76], we study the influence of regularization for ConvNet-based and Transformer-based generators in the full-data regime. We use the whole training set of FFHQ $256 \times 256$, and compare HiT-based variants with StyleGAN2 [26]. The regularization method is bCR [77]. As shown in Table 7, all variants of HiT achieve much larger margins of improvement than StyleGAN2. This confirms the finding [12, 23, 60] that Transformer-based architectures are much more data-hungry than ConvNets in both classification and generative modeling settings, and thus strong data augmentation and regularization techniques are crucial for training Transformer-based generators.

## 5 Conclusion

We present HiT, a novel Transformer-based generator for high-resolution image generation based on GANs. To address the quadratic scaling problem of Transformers, we structure the low-resolution stages of HiT following the design of Nested Transformer and enhance it by the proposed multi-axis blocked self-attention. In order to handle extremely long inputs in the high-resolution stages, we

Table 6: Performance as a function of the number of self-attention stages on ImageNet $128 \times 128$. The attention configuration is defined using the protocol $[a, b]$, where $a$ and $b$ refer to the number of stages in the low-resolution and high-resolution stages of the model, respectively.

| Attention configuration | $[0, 5]$ | $[1, 4]$ | $[2, 3]$ | $[3, 2]$ | $[4, 1]$ |
|---|---|---|---|---|---|
| #params (million) | 61.55 | 66.01 | 67.19 | 67.52 | 67.60 |
| Throughput (images / sec) | 82.67 | 80.88 | 80.22 | 78.06 | 75.33 |
| FID $\downarrow$ | 35.94 | 34.16 | 33.69 | 32.72 | 31.87 |

Table 7: The effectiveness of bCR [77] on both StyleGAN2 and HiT-based variants. $^{\dagger}$ indicates the results of StyleGAN2 are obtained from [26] which uses a lighter-weight configuration of [28].

| + bCR [77] | StyleGAN2 [28][†] | HiT-S | HiT-B | HiT-L |
|---|---|---|---|---|
| ✗ | 5.28 | 6.07 | 5.30 | 5.13 |
| ✓ | 3.91 | 3.06 | 2.95 | 2.58 |
| $\Delta$ FID | 1.37 | 3.01 | 2.35 | 2.55 |

drop self-attention operations and reduce the model into implicit functions. We further improve the model performance by introducing a cross-attention module which plays a role of self-modulation. Our experiments demonstrate that HiT achieves highly competitive performance for high-resolution image generation compared with its ConvNet-based counterparts. For future work, we will investigate Transformer-based architectures for discriminators in GANs and efficient regularization techniques for Transformer-based generators to synthesize ultra high-resolution images.

**Ethics statement.** We note that although this paper does not uniquely raise any new ethical challenges, image generation is a field with several ethical concerns worth acknowledging. For example, there are known issues around bias and fairness, either in the representation of generated images [39] or the implicit encoding of stereotypes [55]. Additionally, such algorithms are vulnerable to malicious use [5], mainly through the development of deepfakes and other generated media meant to deceive the public [67], or as an image denoising technique that could be used for privacy-invading surveillance or monitoring purposes. We also note that the synthetic image generation techniques have the potential to mitigate bias and privacy issues for data collection and annotation. However, such techniques could be misused to produce misleading information, and researchers should explore the techniques responsibly.

## Acknowledgements

We thank Chitwan Saharia, Jing Yu Koh, and Kevin Murphy for their feedback to the paper. We also thank Ashish Vaswani, Mohammad Taghi Saffar, and the Google Brain team for research discussions and technical assistance. This research has been partially funded by the following grants ARO MURI 805491, NSF IIS-1793883, NSF CNS-1747778, NSF IIS 1763523, DOD-ARO ACC-W911NF, NSF OIA-2040638 to Dimitris N. Metaxas.

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
