# Improved Transformer for High-Resolution GANs: Supplementary Material

**Long Zhao**[1]    **Zizhao Zhang**[2]    **Ting Chen**[3]    **Dimitris N. Metaxas**[1]    **Han Zhang**[3]

[1]Rutgers University    [2]Google Cloud AI    [3]Google Research

## 1   Appendix

We provide more architecture and training details of the proposed HiT as well as additional experimental results to help better understand our paper.

### 1.1   Attention Module

The proposed multi-axis blocked self-attention module and cross-attention module follow the conventional design of attention modules presented in Transformers [4, 19], which consist of layer normalization (LN) [1] layers, feed-forward networks (we denoted as MLPs), and residual connections. To be specific, after we add trainable positional embedding vectors [17] $p$ to the input $x$, an attention module updates the input $x$ following the rule:

$$
\begin{aligned}
y &= x + \text{Attention}(x', x', x'), \text{ where } x' = \text{LN}(x) \\
x &= y + \text{MLP}(\text{LN}(y)),
\end{aligned}
\tag{1}
$$

where MLP is a two-layer network: $\max(0, xW_1 + b)W_2 + b$, and $\text{Attention}(Q, K, V)$ denotes the query-key-value attention [2]. Replacing $\text{Attention}(Q, K, V)$ in Equation (1) with the blocking operation and multi-axis self-attention leads to the proposed multi-axis blocked self-attention module, while using the latent embedding as $K$ and $V$ results in the proposed cross-attention module. It is also obvious that if we remove the attention operation in Equation (1), it can be rewritten as $f(p; x)$, where $p$ represents the coordinate of each element in the input and $x$ encodes the condition. Hence, if we omit normalization operations, the function $f(p; x)$ can be viewed as a form of conditional implicit neural functions [3, 12].

To further improve the model performance and efficiency, we make two modifications to the vanilla attention module in [4]. First, we replace the layer normalization (LN) [1] with batch normalization (BN) [7] for image generation. We find BN can not only achieve better IS and FID scores than LN, but also stabilize the training process. Second, we replace the multi-head attention (MHA) [19] with multi-query attention (MQA) [14]. MHA consists of multiple attention layers (heads) in parallel with different linear transformations on the queries, keys, values and outputs. MQA is identical except that the different heads share a single set of keys and values. We observe that incorporating MQA does not lead to degradation in performance but can improve the computational speed of the model. We report detailed results in Table 1 on ImageNet $128 \times 128$.

Table 1: Comparison with different architectural components of the proposed HiT on the ImageNet $128 \times 128$ dataset. **Left**: Layer normalization (LN) [1] and batch normalization (BN) [7]. **Right**: multi-head attention (MHA) [19] and multi-query attention (MQA) [14].

| Method | FID ↓ | IS ↑ | Method | Throughput ↑ | FID ↓ | IS ↑ |
|---|---|---|---|---|---|---|
| HiT w/ LN [1] | 38.96 | 18.18 | HiT w/ MHA [19] | 74.73 | 32.06 | 20.92 |
| HiT w/ BN [7] | **31.87** | **21.32** | HiT w/ MQA [14] | **75.33** | **31.87** | **21.32** |

35th Conference on Neural Information Processing Systems (NeurIPS 2021).

Table 2: Detailed architecture specifications of HiT-{S, B, L} on CelebA-HQ [9] and FFHQ [10] for the output resolutions of $256 \times 256$ and $1024 \times 1024$. We denote cross-attention modules as $(\cdot)$, multi-axis blocked attention modules as $\{\cdot\}$, and MPLs as $|\cdot|$. We also note that stages 1 to 4 are low-resolution stages and 5 to 8 are high-resolution stages.

| Output resolution | | $256 \times 256$ | | | $1024 \times 1024$ |
|---|---|---|---|---|---|
| | Input size | HiT-S | HiT-B | HiT-L | HiT-B |
| Stage 1 | 8 | (dim 512, head 16) × 1 
 $\left\{\begin{array}{l}\text{block sz. } 4\times4 \\ \text{dim 512, head 16}\end{array}\right\} \times 2$ 
 pixel shuffle, 256-d | (dim 512, head 16) × 1 
 $\left\{\begin{array}{l}\text{block sz. } 4\times4 \\ \text{dim 512, head 16}\end{array}\right\} \times 2$ 
 pixel shuffle, 512-d | (dim 1024, head 16) × 1 
 $\left\{\begin{array}{l}\text{block sz. } 4\times4 \\ \text{dim 1024, head 16}\end{array}\right\} \times 2$ 
 pixel shuffle, 512-d | (dim 512, head 16) × 1 
 $\left\{\begin{array}{l}\text{block sz. } 4\times4 \\ \text{dim 512, head 16}\end{array}\right\} \times 2$ 
 pixel shuffle, 512-d |
| Stage 2 | 16 | (dim 256, head 8) × 1 
 $\left\{\begin{array}{l}\text{block sz. } 4\times4 \\ \text{dim 256, head 8}\end{array}\right\} \times 2$ 
 pixel shuffle, 128-d | (dim 512, head 8) × 1 
 $\left\{\begin{array}{l}\text{block sz. } 4\times4 \\ \text{dim 512, head 8}\end{array}\right\} \times 2$ 
 pixel shuffle, 256-d | (dim 512, head 8) × 1 
 $\left\{\begin{array}{l}\text{block sz. } 4\times4 \\ \text{dim 512, head 8}\end{array}\right\} \times 2$ 
 pixel shuffle, 256-d | (dim 512, head 8) × 1 
 $\left\{\begin{array}{l}\text{block sz. } 4\times4 \\ \text{dim 512, head 8}\end{array}\right\} \times 2$ 
 pixel shuffle, 256-d |
| Stage 3 | 32 | (dim 128, head 4) × 1 
 $\left\{\begin{array}{l}\text{block sz. } 8\times8 \\ \text{dim 128, head 4}\end{array}\right\} \times 1$ 
 pixel shuffle, 64-d | (dim 256, head 4) × 1 
 $\left\{\begin{array}{l}\text{block sz. } 8\times8 \\ \text{dim 256, head 4}\end{array}\right\} \times 2$ 
 pixel shuffle, 128-d | (dim 256, head 4) × 1 
 $\left\{\begin{array}{l}\text{block sz. } 8\times8 \\ \text{dim 256, head 4}\end{array}\right\} \times 2$ 
 pixel shuffle, 128-d | (dim 256, head 4) × 1 
 $\left\{\begin{array}{l}\text{block sz. } 8\times8 \\ \text{dim 256, head 4}\end{array}\right\} \times 2$ 
 pixel shuffle, 128-d |
| Stage 4 | 64 | (dim 64, head 4) × 1 
 $\left\{\begin{array}{l}\text{block sz. } 8\times8 \\ \text{dim 64, head 4}\end{array}\right\} \times 1$ 
 pixel shuffle, 32-d | (dim 128, head 4) × 1 
 $\left\{\begin{array}{l}\text{block sz. } 8\times8 \\ \text{dim 128, head 4}\end{array}\right\} \times 2$ 
 pixel shuffle, 64-d | (dim 128, head 4) × 1 
 $\left\{\begin{array}{l}\text{block sz. } 8\times8 \\ \text{dim 128, head 4}\end{array}\right\} \times 2$ 
 pixel shuffle, 128-d | (dim 128, head 4) × 1 
 $\left\{\begin{array}{l}\text{block sz. } 8\times8 \\ \text{dim 128, head 4}\end{array}\right\} \times 2$ 
 pixel shuffle, 64-d |
| Stage 5 | 128 | (dim 32, head 4) × 1 
 \|dim 32\| × 1 
 pixel shuffle, 32-d | (dim 64, head 4) × 1 
 \|dim 64\| × 1 
 pixel shuffle, 64-d | (dim 128, head 4) × 1 
 \|dim 128\| × 2 
 pixel shuffle, 128-d | (dim 64, head 4) × 1 
 \|dim 64\| × 1 
 pixel shuffle, 64-d |
| Stage 6 | 256 | (dim 32, head 4) × 1 
 \|dim 32\| × 1 
 linear, 3-d | (dim 64, head 4) × 1 
 \|dim 64\| × 1 
 linear, 3-d | (dim 128, head 4) × 1 
 \|dim 128\| × 2 
 linear, 3-d | (dim 64, head 4) × 1 
 \|dim 64\| × 1 
 pixel shuffle, 32-d |
| Stage 7 | 512 | | | | (dim 32, head 4) × 1 
 \|dim 32\| × 1 
 pixel shuffle, 32-d |
| Stage 8 | 1024 | | | | (dim 32, head 4) × 1 
 \|dim 32\| × 1 
 linear, 3-d |

## 1.2 Detailed Architectures

The detailed architecture specifications of the proposed HiT on CelebA-HQ [9] and FFHQ [10] are shown in Table 2, where an input latent code of 512 dimensions is assumed for all architectures. "pixel shuffle" indicates the pixel shuffle operation [15] which results in an upsampling of the feature map by a rate of two. "256-d" denotes a linear layer with an output dimension of 256. "block sz. $4 \times 4$" indicates the blocking operation producing non-overlapping feature blocks, each of which has the resolution of $4 \times 4$. On the ImageNet $128 \times 128$ dataset, we use the same architecture of HiT-L as shown in Table 2 except that its last stage for the resolution of $256 \times 256$ is removed. We also reduce the dimension of the input latent code to 256 on this dataset.

## 1.3 Implementation Details

We use Tensorflow for implementation. We provide the detailed description about the generative process of the proposed HiT in Algorithm 1. The Tensorflow code for computing the proposed multi-axis attention is shown in Algorithm 2. Our code samples use einsum notation, as defined in Tensorflow, for generalized contractions between tensors of arbitrary dimension. The pixel shuffle [15] operation is implemented by `tf.nn.depth_to_space` with a block size of two. Feature blocking and unblocking can be implemented by `tf.nn.space_to_depth` and `tf.nn.depth_to_space` with reshape operations, respectively. See Algorithm 3 for more details about blocking and unblocking.

## 1.4 Objectives of GANs

As stated in the main paper, our model is trained with a standard non-saturating logistic GAN loss with $R_1$ gradient penalty [13]. In the original GAN formulation [6], the output of the discriminator $D$ is a probability and the cost function for the discriminator is given by the negative log-likelihood

**Algorithm 1** Generation process of HiT.

**Define:** $X_l$ denotes the feature map in the $l$-th stage. $P_X$ is the positional encoding of $X$. Linear($\cdot$) denotes a linear projection function. ReShape$_{8\times8}(\cdot)$ denotes the operation to reshape the input into the output with the spatial resolution of $8 \times 8$. UpSampleNN$_{2\times2}(\cdot)$ denotes nearest neighbour upsampling with the factor of $2 \times 2$. PixelShuffle$_{2\times2}(\cdot)$ means pixel shuffle upsampling with the block size of $2 \times 2$.

**Input:** the latent code $z$, # of low-resolution stages $M$, and # of high-resolution stages $N$.

**Output:** The final image $I$ with the target resolution.

$I \leftarrow \mathbf{0}$          *# initialize the output image to zeros*

$Z \leftarrow \text{ReShape}_{8\times8}(\text{Linear}(z)) + P_Z$     *# create the latent embedding with positional encoding*

$X_0 \leftarrow \text{ReShape}_{8\times8}(\text{Linear}(z))$     *# create the initial feature map from the input latent code*

**for** $0 \leq l < M + N$ **do**

    $X_l \leftarrow X_l + P_{X_l}$     *# add positional encoding to the feature map of the l-th stage*

    $X_l \leftarrow \text{MultiQueryAttention}(X_l, Z, Z)$     *# perform cross-attention – $X_l$ as Q and Z as K, V*

    **if** $l < M$ **then**

       $X_l \leftarrow \text{Block}(X_l)$     *# block the feature map into patches in low-resolution stages*

       $X_l \leftarrow \text{MultiAxisAttention}(X_l, X_l, X_l)$     *# perform multi-axis self-attention – $X_l$ as Q, K, and V*

       $X_l \leftarrow \text{UnBlock}(X_l)$     *# unblock non-overlapping patches to the feature map*

    **else**

       $X_l \leftarrow \text{MLP}(X_l)$     *# use only MLP in high-resolution stages*

       $I \leftarrow \text{UpSampleNN}_{2\times2}(I) + \text{RGB}(X_l)$     *# upsample and sum the results to create the image*

    **end if**

    $X_{l+1} \leftarrow \text{Linear}(\text{PixelShuffle}_{2\times2}(X_l))$     *# upsample and produce the input for the next stage*

**end for**

---

**Algorithm 2** Tensorflow code implementing Multi-Axis Attention.

```python
def MultiAxisAttention(X, Y, W_q, W_k, W_v, W_o):
  """Multi-Axis Attention.
  X and Y are blocked feature maps where m is # of patches and n is patch sequence length.
  b is batch size; d is channel dimension; h is number of heads; k is key dimension; v is
      value dimension.
  Args:
    X: a tensor used as query with shape [b, m, n, d]
    Y: a tensor used as key and value with shape [b, m, n, d]
    W_q: a tensor projecting query with shape [h, d, k]
    W_k: a tensor projecting key with shape [d, k]
    W_v: a tensor projecting value with shape [d, v]
    W_o: a tensor projecting output with shape [h, d, v]
  Returns:
    Z: a tensor with shape [b, m, n, d]
  """
  Q = tf.einsum("bmnd,hdk->bhmnk", X, W_q)
  Q1, Q2 = tf.split(Q, num_or_size_splits=2, axis=1)
  K = tf.einsum("bmnd,dk->bmnk", Y, W_k)
  V = tf.einsum("bmnd,dv->bmnv", Y, W_v)
  # Compute dilated attention along the second axis of X and Y.
  logits = tf.einsum("bhxyk,bzyk->bhxyz", Q1, K)
  scores = tf.nn.softmax(logits)
  O1 = tf.einsum("bhyxz,bzyv->bhxyv", scores, V)
  # Compute regional attention along the third axis of X and Y.
  logits = tf.einsum("bhxyk,bxzk->bhxyz", Q2, K)
  scores = tf.nn.softmax(logits)
  O2 = tf.einsum("bhxyz,bxzv->bhxyv", scores, V)
  # Combine attentions within heads.
  O = tf.concat([O1, O2], axis=1)
  Z = tf.einsum("bhmnv,hdv->bmnd", O, W_o)
  return Z
```

**Algorithm 3** Tensorflow code implementing feature blocking and unblocking.

```
def Block(X, patch_size=8):
  """Feature blocking.
  Args:
    X: a tensor with shape [b, h, w, d], where b is batch size, h is feature height, w is
      feature width, and d is channel dimension.
    patch_size: an integer for the patch (block) size.
  Returns:
    Y: a tensor with shape [b, m, n, d], where m is # of patches and n is patch sequence
      length.
  """
  _, h, w, d = X.shape
  b = int(patch_size**2)
  Y = tf.nn.space_to_depth(X, patch_size)
  Y = tf.reshape(Y, [-1, h * w // b, b, d])
  return Y

def UnBlock(X, aspect_ratio=1.0):
  """Feature unblocking.
  Args:
    X: a tensor with shape [b, m, n, d], where b is batch size, m is # of patches, n is patch
      sequence length, and d is channel dimension.
    aspect_ratio: a float for the ratio of the feature width to height.
  Returns:
    Y: a tensor with shape [b, h, w, d], where h is feature height and w is feature width.
  """
  _, m, n, d = X.shape
  h = int((m / aspect_ratio)**0.5)
  w = int(h * aspect_ratio)
  patch_size = int(n**0.5)
  Y = tf.reshape(X, [-1, h, w, d * patch_size**2])
  Y = tf.nn.depth_to_space(Y, patch_size)
  return Y
```

of the binary discrimination task of classifying samples as real or fake. Concurrently, the generator $G$ optimizes a cost function that ensures that generated samples have high probability of being real. The corresponding loss functions are defined as:

$$\mathcal{L}_D = -\mathbb{E}_{x \sim P_x}[\log(D(x))] - \mathbb{E}_{z \sim P_z}[\log(1 - D(G(z)))] + \gamma \cdot \mathbb{E}_{x \sim P_x}[\|\nabla_x D(x)\|_2^2], \quad (2)$$

$$\mathcal{L}_G = -\mathbb{E}_{z \sim P_z}[\log(D(G(z)))], \quad (3)$$

where $\gamma$ is the weight of $R_1$ gradient penalty [13] and we set it as 10 in the experiments. $R_1$ gradient penalty penalizes the discriminator from deviating from the Nash Equilibrium via penalizing the gradient on real data alone. We use these adversarial losses throughout our experiments.

### 1.5 Training Details of VQ-HiT

We explore using HiT as a decoder in the vector quantised-variational auto-encoder (VQ-VAE) [18]. We use the same encoder $E$ as the one of VQGAN [5] while replace its ConvNet-based decoder $G$ with the proposed HiT using the HiT-B configuration in Table 2 for producing $256 \times 256$ images.

Apart from the reconstruction loss and GAN loss, we also introduce perceptual loss [8] by using the feature extracted by VGG [16] following [5, 21]. Hence, the training process of VQ-HiT can be formulated as:

$$\mathcal{L}_D = -\mathbb{E}_{x \sim P_x}[\log(D(x))] - \mathbb{E}_{z \sim P_z}[\log(1 - D(G(z)))] + \gamma \cdot \mathbb{E}_{x \sim P_x}[\|\nabla_x D(x)\|_2^2], \quad (4)$$

$$\mathcal{L}_{\text{VAE}} = \|x - G(E(x))\|_2^2 + \lambda_1 \cdot \|F(x) - F(G(E(x)))\|_2^2 - \lambda_2 \cdot \mathbb{E}_{z \sim P_z}[\log(D(G(E(x))))], \quad (5)$$

where $x$ is the input image to be reconstructed, $\lambda_1$ and $\lambda_2$ are the perceptual loss weight and discriminator loss weight, and $F(\cdot)$ denotes the VGG feature extraction model. We set $\lambda_1 = 5 \times 10^{-5}$ and $\lambda_2 = 0.1$ in the experiments. Adam [11] is utilized for optimization with $\beta_1 = 0$ and $\beta_2 = 0.99$. The learning rate is 0.0001 for both the auto-encoder and discriminator. We set the mini-batch size to 256 and train the model for 500,000 iterations.

## 1.6 More Ablation Studies

We implement two additional variants of HiT where the model uses only blocked axial attention and Nested Transformer [20], respectively. The results on the ImageNet $128 \times 128$ dataset are shown in Table 3. We can observe that using only blocked multi-axis attention can still lead to significant performance improvement which demonstrates its effectiveness. We also find that incorporating cross-attention and multi-axis attention as in HiT yields much better performance than vanilla Nested Transformer.

Table 3: Comparison with different variants of the proposed HiT on the ImageNet $128 \times 128$ dataset.

| Method | FID $\downarrow$ | IS $\uparrow$ |
| --- | --- | --- |
| Baseline (INR) | 56.33 | 16.19 |
| Nested Transformer [20] | 42.52 | 18.68 |
| HiT w/ only blocked multi-axis attention | 35.43 | 19.75 |
| HiT w/ only cross-attention | 35.94 | 19.42 |
| HiT (Full model) | **31.87** | **21.32** |

## 1.7 More Qualitative Results

We include more visual results that illustrate various aspects related to generated image quality of HiT. Figure 1 shows uncurated results on the ImageNet $128 \times 128$ dataset. We randomly sample from the ConvNet baseline and the proposed HiT as qualitative examples for a side by side comparison. From the results, we can see that the samples generated by HiT exhibit much more diversities in object category, texture, color, and image background than the ConvNet baseline. Figure 2 shows additional hand-picked synthetic face images illustrating the quality and diversity achievable using our method on the CelebA-HQ $256 \times 256$ and $1024 \times 1024$ dataset.

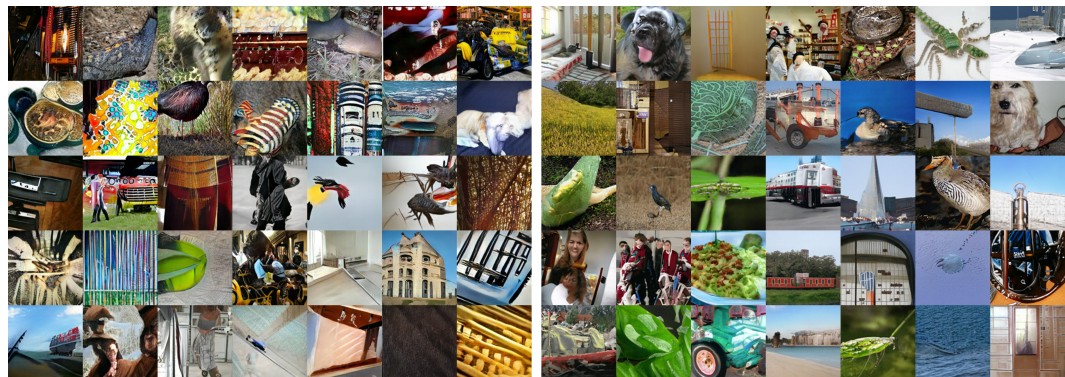

Figure 1: Uncurated ImageNet $128 \times 128$ samples from ConvNet-$R_1$ (**left**, FID: 37.18, IS: 19.55) and the proposed HiT (**right**, FID: 30.83, IS: 21.64). Results are randomly sampled from batches.

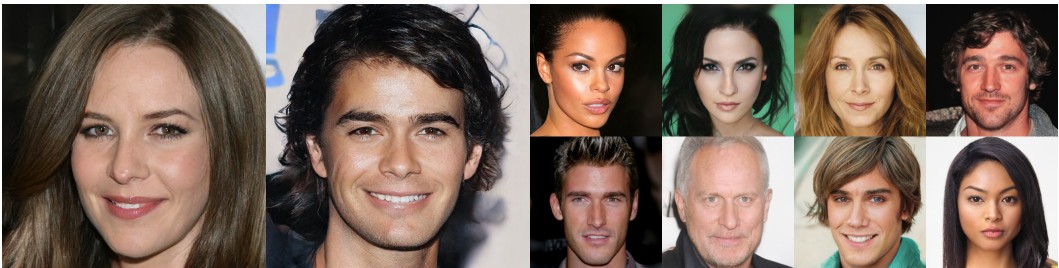

Figure 2: Additional synthetic face images by HiT-B on CelebA-HQ $1024 \times 1024$ and $256 \times 256$.

**Interpolation.** We conclude the visual results with the demonstration of the flexibility of HiT. As well as many other generators, the proposed HiT generators have the ability to interpolate between input latent codes with meaningful morphing. Figure 3 illustrates the synthetic face results on the CelebA-HQ $256 \times 256$ dataset. As expected, the change between the extreme images occurs smoothly and meaningfully with respect to different facial attributes including gender, expression, eye glass, and view angle.

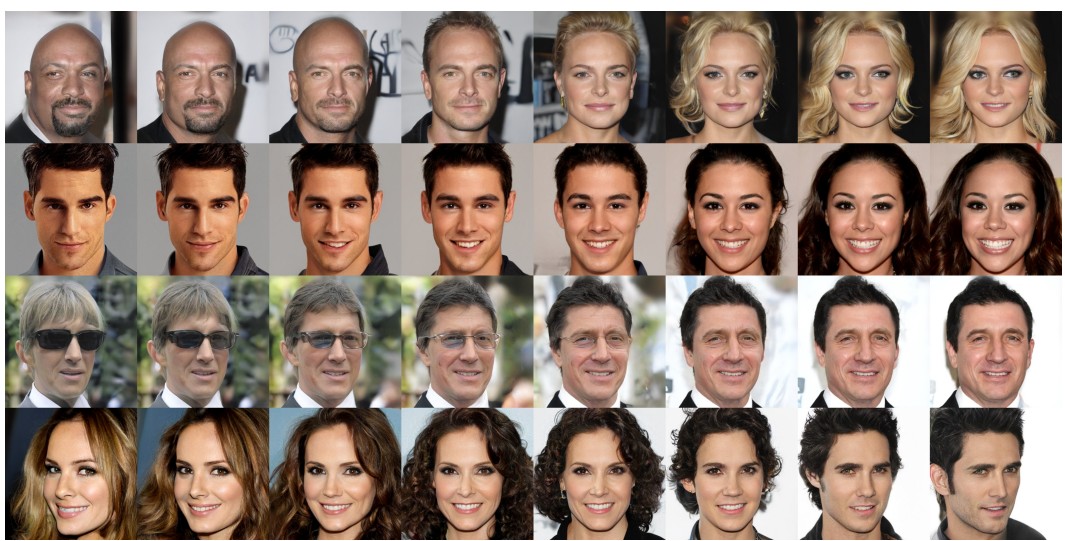

Figure 3: Latent linear morphing on the CelebA-HQ $256 \times 256$ dataset between two synthetic face images – the left-most and right-most ones. HiT-B is able to produce meaningful interpolations of facial attributes in terms of gender, expression, eye glass, and view angle (from **top** to **bottom**).