# OpenReview forum: "Improved Transformer for High-Resolution GANs"
_NeurIPS.cc/2021/Conference — NeurIPS 2021 Poster_

### Official Review · Reviewer_JC42 · 2021-07-13

**Rating:** 4
**Confidence:** 3

**Summary:**

The paper describes a purely attention-based GAN generator architecture where low-resolution deeper layers feature larger-scale (more distant) communication and higher-resolution layers synthesize locally using MLPs,  but still conditioned (indirectly) on the initial latent code.  State-of-the-art results are claimed for 128^2 unconditional ImageNet synthesis (but I believe this claim to be false, see below). In addition, results are shown for CelebA-HQ and FFHQ. While the architecture have some similarities to the near-concurrent work of Hudson and Zitnick (ICML 2021), lack of comparisons should not be held against this paper.

I remain unconvinced by the argumentation and how the results support them. Visually high-quality results are only shown for the simple face image datasets, for which attention mechanisms do not appear to be as necessary as for more complex ones that feature scene compositions. I feel I do not walk away feeling that I learned something.

**Ethical Concerns:**

Nothing but the same things that apply to all image generation work, but the writeup in Sec. 5 is still just a stub.

**Limitations And Societal Impact:**

The required impact statement (end of Sec. 5) does not appear like a serious attempt.

**Main Review:**


The paper argues for an all-attention GAN generator, where the quadratic cost of self-attention is mitigated by restricting explicit long-range dependency modeling to coarse, deep layers.

My largest problem with the paper is that the quest for getting rid of convolutions is not really motivated. We all know metrics like FID and IS only tell a very coarse story, but the submission does not really analyze the properties of the resulting architecture. How does it build the pictures? How is its functionality different from existing main types of model architectures. For instance, compare to the attention map visualizations and composiotionality studies of Hudson and Zitnick that I found very illuminating.

The paper claims a record FID=31.87 and IS=21.32 for unconditional 128x128 ImageNet synthesis. I believe this to be incorrect: the BigBiGAN paper (Donahue and Simonyan, Large Scale Adversarial Representation Learning, NeurIPS 2019) presents an unconditional 128x128 model ("no \mathcal{E} (GAN)" in Table 1) with FID=30.91 and IS=23.56. The claim of "outperforming previous ConvNet-based models with a notable margin" (L235) does not appear to be justified.

Overall, I find image synthesis architectures fascinating, and that the quest for how to best do it is definitely unsolved. While it is interesting that the problem can be solved using all-attention architectures, I am not convinced by arguments that rely solely on metric numbers without analysis of the model's properties ("what makes it tick"). Moreover, the best results here are presented on aligned face datasets that are known to function well on existing architectures. I would be more convinced seeing high-quality results on more complex datasets, supported also by analysis other than FID and IS.

Smaller comments:

The dimensions have been chosen quite poorly in Fig. 2; I find it impossible to understand without consulting the text. The choice of b=2 does not match well with the choice of 4x4 inputs, as dimension=2 is now overloaded.

The authors do not present visualizations of the attention maps and how they relate to the synthesis results.

All synthesis results in Fig. 3 show serious aliasing artifacts. I suspect that the ImageNet dataset has been preprocessed incorrectly when downsizing to 128^2, a common error due to the poor design choices in current deep learning frameworks. (Note that e.g. TORCH.NN.FUNCTIONAL.INTERPOLATE does *not* perform correct filtering when downsampling; comparing its output to e.g. PIL.Image.resize with resample=PIL.Image.BICUBIC or Matlab's imresize will reveal clear differences.)

**Time Spent Reviewing:**

3

---

> ### Author Response · Authors · 2021-08-10
> **Response to Reviewer JC42**
>
> We appreciate the detailed comments from the reviewer. We think the reviewer misunderstood the motivation and goal of this paper. Below are our responses to the points raised by the reviewer. We sincerely hope the reviewer can re-evaluate this paper based on our responses.
>
> 1. __Clarification on the motivations.__ The motivation of this paper is not to get rid of convolutions. Instead, we are motivated by the recent notable learning capabilities shown in attention-based models and aim to build efficient attention-based architectures for high-resolution image generation. Our resulting model is free of convolution because of the design choice we made after balancing the tradeoff between performance and computational cost (see our fifth response to Reviewer 6oDm). Our experimental results show that the proposed HiT can outperform the current convolution-based models which demonstrate the validity of our motivation. We will clarify this point in the final version
>
> 2. __Difference between GANformer and HiT.__ GANformer (Hudson and Zitnick, ICML 2021) and HiT are different in the goal of using attention modules. GANformer utilizes the attention mechanism to model the dependences of objects in a generated scene/image. Instead, HiT explores building efficient attention modules for synthesizing general objects. Our experiments demonstrate that even for simple face image datasets, incorporating the attention mechanism can still lead to performance improvement. We believe our work reconfirms the necessity of using attention modules for general image generation tasks, and more importantly, we present an efficient architecture for attention-based high-resolution generators which might benefit the design of future attention-based GANs. Due to the limited time during the rebuttal period, we will provide quantitative comparisons with GANformer on more complex datasets in the final version.
>
> 3. __Comparison with BigGANs.__ We appreciate the reviewer for pointing out the result of BigGAN. However, we note that the BigGAN model is far larger (more than 160M # of params) than HiT (around 50M # of params), so the results of these two models are not directly comparable. Even though, HiT (31.78 FID) is only slightly shy of BigGAN (30.91 FID) and, more importantly, HiT does outperform previous ConvNet-based models with a similar capacity by a notable margin as shown in Table 1 of the original paper. We believe these results can demonstrate the state-of-the-art performance of HiT. We will present these results in a clearer way in the final version.
>
> 4. __Visualizations of the attention maps.__ Thanks for the suggestion. Since NeurIPS does not allow providing additional visual results in the rebuttal, we will add the visualizations in the final version.
>
> 5. __Aliasing artifacts in Figure 3.__ Thanks for pointing this out. Our model is implemented by Tensorflow, and we use the standard function `tf.image.resize` to downsize the input images. We will double-check this implementation and make sure that all images are preprocessed correctly in our final version.
>
> We will also address the reviewer’s concerns on the clarity of figures and other minor comments in the final version. We thank the reviewer for the feedback and consideration.

---

> ### Author Response · Authors · 2021-09-02
> **Look forward to your updated review**
>
> Dear Reviewer JC42,
>
> Thank you for your detailed and instructive reviews. Reviewers 6oDm and QCfs have responded, and Reviewer 6oDm increased their scores and others remain their scores. We would like to know if our response has resolved your concerns. In our response, we believe we have addressed your main concerns on our motivations, differences with GANformer, and comparisons with BigGANs. In our revision, we plan to add detailed experimental results compared with GANformer and BigGANs, provide visualizations of the attention maps, and address the rest concerns on the clarity of figures and other minor comments. We would really appreciate it if you could update your review given our response.
>
> Thanks again for your time!

---

### Official Review · Reviewer_Uc2m · 2021-07-14

**Rating:** 7
**Confidence:** 5

**Summary:**

This paper has presented HiT, a Transformer-based generator for high-resolution image synthesis based on GANs. Instead of using full attention, this paper has proposed the multi-axis blocked self-attention, which captures local and global dependencies within non-overlapping image blocks in parallel. In addition, HiT introduces the cross-attention mechanism to integrate noise information into multi-stages of the generator. Extensive experiments demonstrate the superiority of the proposed method.

**Limitations And Societal Impact:**

The authors have addressed the limitations and potential negative societal impact.

**Main Review:**

Originality:  This paper presents a new Transformer-based architecture for the generator in GANs. The authors introduce several new techniques to make it work for the Transformer-based generative model.

Quality: This submission is technically sound and most claims are well supported.

Clarity: This paper is clearly written and well organized.

Significance: The experiment results are very important for exploring Transformer-based architecture for generative models.

Weakness:
1. I am curious about what is a good noise injection method in the Transformer-based generative models. Previous well-known methods StyleGAN and StyleGAN2 apply the AdaIN and modulated convolution for noise injection. HiT proposes the cross-attention for noise injection, the authors should conduct experiments to compare cross-attention with these methods, maybe AdaIN and modulated convolution also work for the Transformer-based generative model.

2. Since the Transformer-based architectures are more data-hungry, maybe the authors can conduct experiments on large-scale datasets: LSUN-cats and LSUN-cars and compare the performance with StyleGAN2 without using bCR.

3. Transformer-based architectures have shown effectiveness in classification models, one of the most important reasons is the capability of capturing long-range dependencies. But in HiT, the self-attention module is only applied in low-resolution stages, make the generator fails to capture long-range dependencies in high-resolution stages. The authors may give some explanations for self-attention modules in low-resolution stages are enough in generative high-quality images.

**Time Spent Reviewing:**

8 hours

---

> ### Author Response · Authors · 2021-08-10
> **Response to Reviewer Uc2m**
>
> We thank the reviewer’s positive feedback and detailed recognition of our contribution. We address the questions raised by the reviewer below to improve the clarity of our paper.
>
> 1. __What is a good noise injection method in Transformer-based generators?__ In our initial experiments, we find that compared with cross-attention, using the AdaIN and modulated layers for Transformer-based generators requires much higher memory cost during model training, which usually leads to out-of-memory errors when the model is trained for generating high-resolution images. As a result, related work like ViT-GAN, which uses AdaIN and modulated layers, can only produce images up to the resolution of $64 \times 64$. Hence, we believe cross-attention is a better choice for high-resolution generators based on Transformers. We will clarify this point in the final version.
>
> 2. __Experiments on large-scale datasets.__ We believe our results on ImageNet $128 \times 128$ can demonstrate the performance of the proposed HiT on large-scale datasets. We are not able to finish the experiments on LSUN-cats and LSUN-cars given the limited time during the rebuttal period. We will add the relevant results in the final version.
>
> 3. __Why self-attention modules in low-resolution stages are enough?__ This is because the low-resolution stages of HiT focus on encoding the structures of images which can benefit from capturing long-range dependencies by using self-attention modules. Instead, high-resolution stages generate image local details so long-range dependencies become less important. This can be demonstrated from Table 6 of the original paper, where we can see that the performance gain achieved by self-attention decreases along with the increase of feature resolutions. Therefore, we drop self-attention modules in high-resolution stages to boost the model efficiency of HiT. We will provide a detailed explanation for this part in the final version as suggested.

---

### Official Review · Reviewer_TdeG · 2021-07-14

**Rating:** 8
**Confidence:** 5

**Summary:**

This paper proposes a new transformer-based generator for high-resolution image generation. It addresses the quadratic scaling problem of the attention operator with multi-axis blocked self-attention which considering the within blocks and across blocks attention. The proposed cross-attention for input and intermediate features is also novel. The evaluation is done in ImageNet and FFHQ datasets.  Promising results are provided.

**Limitations And Societal Impact:**

I think the authors have addressed the limitations and potential negative societal impact.

**Main Review:**

This is a well-written paper with carefully designed experiments. The motivation for utilizing the multi-axis blocked self-attention is reasonable. The proposed cross-attention for input and intermediate features is novel and interesting. The authors conduct various experiments to show the effectiveness of the proposed results, both quantitatively and qualitatively.

Line 12: "Hit has a linear computational complexity" is a bit overclaim. Since the full HiT model uses balancing attention between axes and its computational complexity is N*sqrt(N).

Why not apply bCR at 1024x1024 resolution? Is not using bCR the reason why HIT is worse than StyleGAN2 at the 1024x1024 resolution?

**Time Spent Reviewing:**

6

---

> ### Author Response · Authors · 2021-08-10
> **Response to Reviewer TdeG**
>
> We appreciate the very positive reviews and detailed recognition of our contribution. We address the questions raised by the reviewer below to improve the clarity of our paper.
>
> 1. __Computational complexity.__ Thanks for pointing this out. We will revise this part and provide a more detailed analysis of the computational complexity (see our fourth response to Reviewer QCfs) of the full HiT model as suggested in the final version.
>
> 2. __Why not apply bCR at $1024 \times 1024$ resolution?__ In the experiment, we find that applying bCR at $1024 \times 1024$ resolution to HiT will lead to non-convergence of the model. This should be the reason why HiT is worse than StyleGAN2 at this resolution. We believe how to apply regularization techniques like bCR to Transformer-based architectures for synthesizing ultra-high-resolution images triggers a new research direction and will explore it in our future work. We will clarify this part in the final version.

---

### Official Review · Reviewer_QCfs · 2021-07-15

**Rating:** 5
**Confidence:** 3

**Summary:**

To address the quadratic complexity of the self-attention operation, this paper proposes a new Transformer-based generator for high-resolution image generation based on GANs, denoted as HiT. In the low-resolution stage, the authors propose a multi-axis blocked self-attention. In the high-resolution stage, they keep multi-layer perceptrons reminiscent of the implicit neural function. Extensive experiments demonstrate the effectiveness of the proposed method.

**Ethics Review Area:**

["I don’t know"]

**Limitations And Societal Impact:**

Yes

**Main Review:**

1. Some motivations should be highlighted. What are the issues of directly using ViT into GAN? In this paper, the authors do not directly apply ViT in GAN. However, there are some papers that propose ViT-based GAN methods. In the low-resolution stage, why replace standard global self-attention. In the high-resolution stage, why drop self-attention in the proposed method?

2. Some related work should be discussed in the paper. There are some papers [1, 2] that propose ViT-based GAN methods. Could you please discuss the difference between them? In addition, one recent paper [3] proposes new attention to generate high-resolution images. Could you please discuss it in the related work?

    [1] TransGAN: Two Pure Transformers Can Make One Strong GAN, and That Can Scale Up, arxiv 2021

    [2] ViTGAN: Training GANs with Vision Transformers, arxiv 2021

    [3] Video Super-Resolution Transformer, arxiv 2021

3. The novelty of the proposed method should be highlighted.  The proposed method seems to combine the Nested Transformer and the Axial attention. The authors should highlight that such a combination is non-trivial.

4. Some technical details are not clear. In Figure 1, what is block $B_i$? Are the two $B_0$ the same in the figure? In Line 154, why $b \times b \sim H/b \times W/b$? In Lines 157 and 174, could you please give a more detailed analysis of complexity?

5. The experiments can be improved. In Table 3, it would be better to compare with BigGANs [4].  In the ablation study,  the authors should compare with the Nested Transformer.

[4] Large Scale GAN Training for High Fidelity Natural Image Synthesis, ICLR 2019


**Time Spent Reviewing:**

5 hours

---

> ### Author Response · Authors · 2021-08-10
> **Response to Reviewer QCfs**
>
> We thank the reviewer’s instructive comments. Below are our responses to the points raised by the reviewer. We sincerely hope these will help improve the clarity of the paper and help the reviewer to finalize the judgment.
>
> 1. __Highlighting the motivations.__ The issue of directly using ViT into GAN is that the very high computational cost of the vanilla self-attention used by ViT makes the model fail to generate ultra-high-resolution images. That’s the reason why ViT-based GANs like [1] and [2] can only produce images up to the resolution of $64 \times 64$. This motivates us to design a more efficient Transformer-based generator, where we use blocked multi-axis attention in low-resolution stages and drop self-attention in high-resolution stages of HiT. We will highlight these motivations in the final version.
>
> 2. __Related work.__ Thanks for pointing out the related work. We note that TransGAN [1] has been already discussed in Section 2 of the original paper. Both [2] and [3] are available after the NeurIPS submission so they are not included in the current version. [2] improves [1] by using modulated layers as in StyleGAN2 and more advanced training schema, but they both fail to generate high-resolution images (e.g., $1024 \times 1024$) due to the high computational cost of ViT-based self-attention. [3] also presents local self-attention within image blocks but it does not perform self-attention across blocks as in HiT. We will discuss them in the final version as suggested.
>
> 3. __Highlighting the novelty.__ Thanks for the suggestion. Our novelty lies in the following three aspects: (1) a two-stage attention-based framework combining (2) and (3) which can generate ultra-high-resolution images; (2) blocked multi-axis attention that improves the axial attention; (3) efficient self-modulation with cross-attention. All of them are non-trivial. We will highlight them in the final version.
>
> 4. (1) __In Figure 1, what is block $B_i$? Are the two $B_0$ the same in the figure?__ $B_i$ means the i-th image block (non-overlapping patch) as shown in Figure 2 of the original paper. They are different across stages. (2) __In Line 154, why b×b∼H/b×W/b?__ In order to balance the sequence length between different attention axes, we need to select the value of $b$ so that the block size $b \times b$ and the number of blocks $\frac{H}{b} \times \frac{W}{b}$ should be as close as possible. (3) __In Lines 157 and 174, could you please give a more detailed analysis of complexity?__ In general, performing dot-product attention between two input sequences of length $N$ and $M$ requires a total of $\mathcal{O}(NM)$ computation. When computing the balanced multi-axis attention (Line 157) on an image with the block size of $S$, i.e., $S^2 = N = M$, we perform attention on $S$ sequences of length $S$, which is a total of $\mathcal{O}(S \cdot S^2) = \mathcal{O}(N\sqrt{N})$ computation. In the case of cross-attention (Line 174), since the embedding size $M$ is pre-defined and usually far less than $N$, i.e., $M \ll N$, so the cross-attention operation has linear complexity $\mathcal{O}(N)$.
>
> 5. (1) __Comparison with BigGANs.__ As reported in the BigBiGAN paper, BigGAN achieves 30.91 FID on unconditional ImageNet $128 \times 128$. But we note that the BigGAN model is far larger (more than 160M # of params) than HiT (around 50M # of params), so the results are not directly comparable. Even though, HiT (31.78 FID) is only slightly shy of BigGAN and achieves the state of the art among models with a similar capacity as shown in Table 1 of the original paper. (2) __Comparison with Nested Transformer.__ The table below shows the comparison with Nested Transformer on ImageNet $128 \times 128$. We can see that incorporating cross-attention and multi-axis attention as in HiT yields much better performance than vanilla Nested Transformer. We will add all these results in the final version.
> | Method | FID | IS |
> | ----------------------------- | -------- | -------- |
> | Nested Transformer | 42.52 | 18.68 |
> | HiT | 31.87 | 21.32 |

---

> > ### Comment · Reviewer_QCfs · 2021-09-02
> > **Keep initial rating, but no objection to acceptance**
> >
> > Thank you for your response. The rebuttal provides valuable information, but some concerns are not fully addressed. I mainly concern on the novelty of the proposed method and the ImageNet results. I will keep the initial score, but I have no objection to the acceptance of this paper.

---

### Official Review · Reviewer_6oDm · 2021-07-16

**Rating:** 7
**Confidence:** 4

**Summary:**

The paper proposes a transformer based generative model which shows competitive results on multiple image generation benchmarks. It combines existing techniques of blocked and axial attention to reduce the computational complexity of self-attention mechanism. To further reduce the complexity, the proposed attention mechanism is only used at lower resolution blocks. The paper also employs technique similar to self-modulation to condition image generation process on the input latent code at all levels.


**Limitations And Societal Impact:**

Yes, the paper touched upon the limitations and societal impact of the work in the conclusion.

**Main Review:**

The paper combines two existing techniques of blocked and axial attention in a novel way to create a convolutional free generator. The final proposed architecture shows competitive or better image generation compared to existing state of the art techniques on multiple datasets according to the FID metric.

**Clarity**: The paper is easy to follow and well written.

**Additional comments and questions**:
* The paper proposes cross-attention which is similar to self-modulation technique. It would be good to see an ablation experiment that shows the performance of using self-modulation instead of cross-attention and its added advantage.
* Will the proposed architecture also provide benefit in case of conditional image generation?
* In Table 5, the paper only shows the result with cross-attention+blocked axial attention, It would be interesting to see the benefit when only blocked axial attention is employed without using cross-attention.
* In Table 6, does having blocked axial attention for all layers (i.e. configuration [5,0]) leads to high memory requirement as it is not shown?
* Though the paper proposes convolutional free generator and therefore has MLPs at the higher resolution blocks, but will having convolutional blocks instead of MLPs at later blocks provide any benefit?
Does the ConvNetR1 baseline includes self-attention layer at one the generator blocks similar to self-attention GANs?

**Related work missing**: One of the relevant related work "Self-supervised gans via auxiliary rotation loss." is missing and would be great to see a fair performance comparison with this as the baseline. This above work shows 23.6 FID metric on unconditional image generation on ImageNet 128x128 dataset which is better than the score reported by the proposed technique.

####################
Update:

Thanks for addressing all the comments in the rebuttal. I am adjusting my score based on the answers.

Additional comments: Are the results on ImageNet dataset with HiT-S architecture? If possible, it would be interesting to compare HiT-L (with similar #params as BigGANs) with BigGAN for ImageNet dataset in the final draft of paper.
After reading other reviewer’s comment, I agree about including attention visualization in the paper.



**Time Spent Reviewing:**

6-7

---

> ### Author Response · Authors · 2021-08-10
> **Response to Reviewer 6oDm**
>
> We thank the reviewer’s detailed comments and recognition of our novelty and state-of-the-art performance. Below are our responses to the points raised by the reviewer. We hope these will help improve the clarity of the paper.
>
> 1. __Comparison with self-modulation.__ Thanks for the suggestion. The table below shows the comparison between HiT models using cross-attention and self-modulation on ImageNet $128 \times 128$, respectively. We can see that the proposed cross-attention achieves better performance since the attention mechanism is more powerful than the normalization form used in self-modulation. We will add the result in the final version.
> | Method | FID | IS |
> | ----------------------------- | -------- | -------- |
> | HiT w/ self-modulation | 33.15 | 20.86 |
> | HiT w/ cross-attention | 31.87 | 21.32 |
>
> 2. __Will the proposed architecture also provide benefit in case of conditional image generation?__ Yes, we believe so since we do not make any specific assumption about the model input during the architecture design of HiT, and thus it should also be able to benefit conditional image generation. But adding conditions to HiT is non-trivial and we will explore it in our future work.
>
> 3. __Comparison with the model using only blocked axial attention.__ Thanks for the suggestion. The table below shows the model performance when only blocked multi-axis attention is used on ImageNet $128 \times 128$. We can observe that using only blocked multi-axis attention can still lead to significant performance improvement which demonstrates its effectiveness. We will include these results in the final version.
> | Method | FID | IS |
> | ----------------------------- | -------- | -------- |
> | Baseline (INR) | 56.33 | 16.19 |
> | HiT w/ only blocked multi-axis attention | 35.43 | 19.75 |
> | HiT w/ only cross-attention | 35.94 | 19.42 |
> | HiT (Full model) | 31.87 | 21.32 |
>
> 4. __Does having blocked axial attention for all layers (i.e. configuration [5,0]) lead to high memory requirement as it is not shown?__ Yes, we do not show this configuration since it requires high memory cost but no additional improvement can be achieved over the configuration [4, 1].
>
> 5. (1) __Will having convolutional blocks instead of MLPs at later blocks provide any benefit?__ In our initial experiments, we do not observe notable performance improvement if we use convolutional blocks instead of MLPs in high-resolution stages, while convolutional operations lead to much higher memory and computational cost. Thus we chose MLPs after balancing the tradeoff between performance and cost. (2) __Does the ConvNetR1 baseline include self-attention layers similar to self-attention GANs?__ Yes, it follows the same design of self-attention GANs. We will clarity these in the final version.
>
> 6. __Comparison with "Self-supervised gans via auxiliary rotation loss" (SS-GAN).__ Thanks for pointing out the missing related work. We note that the SS-GAN model achieving 23.6 FID uses the BigGAN’s architecture which is far larger (more than 160M # of params) than HiT (around 50M # of params). But when compared to the SS-GAN model having a similar model capacity (Table 1 of the SS-GAN paper) with HiT, HiT (31.87 FID) still outperforms SS-GAN (43.87 FID) by a large margin. We will include these comparisons in the final version.

---

### Decision · Program_Chairs · 2021-09-27

**Decision:**

Accept (Poster)

**Comment:**

The paper proposes a transformer-based generative model that shows better results on multiple standard benchmarks. The model includes multi-axis blocked self-attention at early stages and uses MLP for late stages. The reviewers appreciate the idea of a transformer-based generator, clear writing, and experimental results. There are also concerns about the STOA results on ImageNet (compared to BigBiGAN) and the motivation and advantage of transformer-based architecture compared to ConvNet. The rebuttal addressed some of the concerns. Overall, the significance of the work outweighs the limitations. I recommend accepting the work. The authors are encouraged to carefully examine the downsampling functions for 128x128 Imagenet (use antialias=True in TF, for example) and discuss the latest results (e.g., BigBiGAN) in ImageNet experiments.